# Cerebrovascular Disease Hospitalization Rates in End-Stage Kidney Disease Patients with Kidney Transplant and Peripheral Vascular Disease: Analysis Using the National Inpatient Sample (2005–2019)

**DOI:** 10.3390/healthcare12040454

**Published:** 2024-02-10

**Authors:** Tyler John Canova, Rochell Issa, Patrick Baxter, Ian Thomas, Ehab Eltahawy, Obi Ekwenna

**Affiliations:** 1College of Medicine and Life Sciences, The University of Toledo, Toledo, OH 43614, USA; 2Department of Internal Medicine, Cleveland Clinic Foundation, Cleveland, OH 44195, USA; issaroch234@gmail.com; 3Schar School of Policy and Government, George Mason University, Fairfax, VA 22030, USA; pbaxter2@gmu.edu; 4Department of Nephrology & Transplant, Mount St. John’s Medical Center, St. John’s, Antigua and Barbuda; samohtnai1111@gmail.com; 5Department of Cardiovascular Medicine, The University of Toledo Medical Center, Toledo, OH 43614, USA; ehab.eltahawy2@utoledo.edu; 6Department of Urology & Transplant, The University of Toledo Medical Center, Toledo, OH 43614, USA; obinna.ekwenna@utoledo.edu

**Keywords:** cerebrovascular hemorrhage, cerebrovascular ischemia, end-stage kidney disease, kidney transplant, peripheral vascular disease

## Abstract

Individuals with end-stage kidney disease (ESKD) face higher cerebrovascular risk. Yet, the impact of peripheral vascular disease (PVD) and kidney transplantation (KTx) on hospitalization rates for cerebral infarction and hemorrhage remains underexplored. Analyzing 2,713,194 ESKD hospitalizations (2005–2019) using the National Inpatient Sample, we investigated hospitalization rates for ischemic and hemorrhagic cerebrovascular diseases concerning ESKD, PVD, KTx, or their combinations. Patients hospitalized with cerebral infarction due to thrombosis/embolism/occlusion (CITO) or artery occlusion resulting in cerebral ischemia (AOSI) had higher rates of comorbid ESKD and PVD (4.17% and 7.29%, respectively) versus non-CITO or AOSI hospitalizations (2.34%, *p* < 0.001; 2.29%, *p* < 0.001). Conversely, patients hospitalized with nontraumatic intracranial hemorrhage (NIH) had significantly lower rates of ESKD and PVD (1.64%) compared to non-NIH hospitalizations (2.34%, *p* < 0.001). Furthermore, hospitalizations for CITO or AOSI exhibited higher rates of KTx and PVD (0.17%, 0.09%, respectively) compared to non-CITO or AOSI hospitalizations (0.05%, *p* = 0.033; 0.05%, *p* = 0.002). Patients hospitalized with NIH showed similar rates of KTx and PVD (0.04%) versus non-NIH hospitalizations (0.05%, *p* = 0.34). This nationwide analysis reveals that PVD in ESKD patients is associated with increased hospitalization rates with cerebral ischemic events and reduced NIH events. Among KTx recipients, PVD correlated with increased hospitalizations for ischemic events, without affecting NIH. This highlights management concerns for patients with KTx and PVD.

## 1. Introduction

Cardiovascular disease remains the leading cause of morbidity and mortality among patients with end-stage renal disease (ESKD) [1]. With nearly 786,000 people in the United States living with ESKD [2], including 29% having undergone kidney transplantation, understanding the comorbid conditions associated with this disease becomes of paramount importance [2,3]. Despite advancements in dialysis, the 24-month and 5-year survival rates for ESKD patients in the United States remains low [4], and cardiovascular disease remains the major contributor to mortality in this patient population [5].

Patients with ESKD are more likely to develop major cardiovascular diseases and syndromes, such as coronary artery disease (CAD), cerebral infarction, and peripheral vascular disease (PVD), compared to those without ESKD [6,7]. PVD, in particular, has been identified as an independent determinant of all-cause mortality and cardiovascular events in ESKD patients, and is present in more than 24% of patients with an estimated glomerular filtration rate (eGFR) < 60 mL/min/1.73 m^2^ [8,9]. Furthermore, PVD is a marker for systemic atherosclerosis and an independent risk factor for myocardial infarction and stroke [10,11].

Despite extensive research, the precise mechanistic relationship between ESKD and the resultant vascular dysfunction seen in PVD and other cardiovascular syndromes remains unclear. The pathogenesis is thought to involve multiple factors, including inflammation, oxidative stress, vascular calcification, uremic toxin accumulation, volume overload, and anemia [12]. Atherosclerotic plaques found in patients with ESKD are characterized by increased calcification and inflammation, contributing to plaque vulnerability and heightened cardiovascular risk [12]. 

The risk of cardiovascular events is further exacerbated in patients who undergo renal transplantation, as they have a 45% higher risk of myocardial infarction compared to patients without a renal transplant [13]. Additionally, transplant patients who experience a myocardial infarction are independently associated with a 15% higher risk of all-cause death compared to patients with myocardial infarction and no renal transplant [13]. Although renal transplantation improves survival and quality of life, it does not fully resolve the metabolic derangements caused by ESKD [5]. Traditional cardiovascular risk factors, including hypertension, dyslipidemia, diabetes mellitus, coexisting renal impairment, and left-ventricular hypertrophy [14] typically persist or worsen post-transplantation, while immunosuppressive therapy required after transplantation further contributes to the increased risk of cardiovascular events in these patients [15,16]. 

Given the increased risk of systemic cardiovascular disease in patients with ESKD, including those who undergo renal transplantation, it becomes crucial to further explore the characteristics of patients who develop additional cardiovascular events, such as cerebral infarction. PVD often precedes other cardiovascular events in this population, making its identification an important marker for systemic disease progression. Furthermore, patients with PVD frequently receive single- or, in many cases, dual-anti-platelet therapy due to the presence of PVD, along with potential coexisting cardiovascular conditions [17]. 

For those patients with PVD undergoing renal transplantation, these anti-platelet medications are commonly paused during the perioperative phase [18], which may expose these patients to an increased risk of vascular events. For instance, there is a potential for cerebral infarction due to heightened coagulation during the early postoperative period, stemming from systemic inflammation [19]. Alternatively, there is also an increased risk of major hemorrhage for kidney transplant patients in general [20]. The balance between the initiation and withdrawal of anti-platelet therapy prior to, during the perioperative period, and post-transplant further complicates the management of patients with ESKD or a kidney transplant and concurrent PVD. 

While numerous studies have individually investigated ESKD, PVD, and cerebral infarction, and intracranial hemorrhage [1,6,7,9,12,14,21], there is a sparsity of literature specifically examining patients with ESKD or kidney transplants who develop cerebral infarction or hemorrhage in the presence of PVD. Despite the high prevalence of these concurrent conditions, there is a need to evaluate the specific subset of patients with ESKD or kidney transplants and PVD who experience cerebral infarction or hemorrhage. 

In this study, we utilize the National Inpatient Sample (NIS) to evaluate the hospitalization rates of various cerebral infarction or hemorrhage events in hospitalized patients with ESKD, PVD, kidney transplant, or a combination of these conditions.

## 2. Materials and Methods

### 2.1. Objectives and Outcomes 

The primary objective of this study is to explore the impact of PVD and KTx on hospitalization rates for cerebral infarction and hemorrhage in patients with ESKD. The primary outcome was to determine whether the presence of PVD and the history of KTx were associated with a statistically significant difference in the rates of hospitalization for cerebral infarction and hemorrhage among patients with end-stage kidney disease (ESKD).

### 2.2. Data, Data Sources, and Sample Details 

The HCUP-NIS dataset underlies all planned analysis. Available through the Agency for Healthcare Research and Quality (AHRQ), the NIS is the largest publicly available all-payer inpatient healthcare database available to researchers. The dataset annually approximates a 20% stratified sample of all episode-level hospitalizations from U.S. community hospitals, and is primarily designed to produce U.S. regional and national estimates of inpatient utilization, access, cost, quality, and outcomes [22]. Available from 1988 to 2019, the complete dataset represents nearly 85% of all national hospitals over that time period. Each year includes around 7 million hospital visits, which represent nearly 35 million hospitalizations once correctly weighted [23].

The utilized dataset consists of a 2005–2019 subset of the NIS database, sorting through 113,032,831 discharges to match each performance measure to a set of corresponding ICD-9-CM from 2005 to 2014, or ICD-10-CM diagnosis codes from 2016 to 2019, summarized in Appendix A. Each ESKD patient in the dataset with at least one matching diagnosis code is compiled into a working dataset (*n* = 2,713,194). Additionally, all other listed diagnoses and procedures, discharge status, patient demographics, and charges associated to each discharge are included in the initial compiled dataset. Elixhauser comorbidity indices are then calculated for each recorded instance [24].

### 2.3. Variables 

Figure 1 depicts a PRISMA flow diagram representing the ESKD hospitalizations included in this study. The inclusion criteria for our study strictly focuses on all inpatient hospitalizations for ESKD patients, including those who underwent a KTx and/or were diagnosed with PVD, as compared to ESKD patients without these specific conditions, within the timeframe of 2005–2019. Appendix A include specific ICD9/10 codes utilized in this study, and more information regarding the specific ICD9/10 code inclusion/exclusion criteria can be found at www.ICD10data.com (accessed on 25 January 2024) [25].

The independent variables included patients with ESKD alone, PVD and ESKD, patients with a KTx alone, or patients with KTx and PVD. Patients categorized as KTx encompass both those with and without PVD. Likewise, patients categorized as ESKD alone encompass both those with and without PVD. Three key conditions of interest used as the dependent variables are included in Appendix A. These included (1) Cerebral infarction due to thrombosis, embolism, occlusion, and stenosis (CITO); (2) Artery occlusion and stenosis resulting in cerebral ischemia (AOSI); and (3) Nontraumatic intracranial hemorrhage (NIH). Each of these key diagnoses of interest corresponds with a set of ICD-10-CM or corresponding ICD-9-CM diagnosis codes, summarized in Appendix A.

### 2.4. Analysis 

Retrospective descriptive analysis and multivariate logistic regressions were performed to examine how the probability of each condition of interest varies between ESKD patients, kidney transplant recipients, and if they additionally had PVD. Descriptive analysis was performed using a *t*-test for continuous variables and Chi-squared or Fisher’s exact tests for categorical variables, depending on the sample size and the distribution of the variables included. Logistic regressions were performed using the three key conditions of interest as the dependent variables, with KTx and PVD as the key independent variables [22,26]. A significance level of 0.05 was employed to assess statistical significance. Covariates included a series of sociodemographic proxies with complete variable details reported in Appendix A.

## 3. Results

### 3.1. Descriptive Statistics, Baseline Characteristics, and Frequency Distribution of ESKD and Kidney Transplant Hospitalizations with and without PVD

The study identified a total of 2,713,194 hospitalizations with ESKD from 2005 to 2019. In Table 1, we present a detailed breakdown of the three key conditions of interest, along with their respective frequencies for ESKD hospitalizations without PVD (*n* = 2,584,262), ESKD hospitalizations with PVD (*n* = 62,005), KTx hospitalizations without PVD (*n* = 65,521), and KTx hospitalizations with PVD (*n* = 1406). Additionally, Table 1 reports the frequencies of selected risk factors and comorbidities, such as essential hypertension, obesity, nicotine dependence (cigarettes), type II diabetes mellitus, atrial fibrillation, and hyperlipidemia, for various patient groups. Moreover, we provide the relative frequencies of specific demographic information, including age, race, primary expected payer, median household income, and patient location for each corresponding patient group.

### 3.2. Baseline Characteristics for All ESKD Patients with PVD versus Patients without PVD

In Table 2, we delineate ESKD hospitalizations with peripheral vascular disease (PVD) (*n* = 63,411) and ESKD hospitalizations without PVD (*n* = 2,649,783), analyzing the three key conditions of interest, as well as assessing associated risk factors, comorbidities, and demographic information, including age, race, primary expected payer, median household income, and patient’s location.

Among ESKD patients hospitalized with PVD, a higher likelihood of hospitalization with CITO (0.15%) or AOSI (3.07%) was observed compared to individuals without PVD (0.08%, *p* < 0.001; 0.93%, *p* < 0.001, respectively). Conversely, ESKD hospitalizations with PVD were less likely to be associated with NIH (0.44%) than those without PVD (0.63%, *p* < 0.001) (Table 2).

Furthermore, ESKD patients with PVD exhibited a significantly higher prevalence of all examined risk factors, including essential hypertension, obesity, nicotine dependence (cigarettes), type II diabetes mellitus, atrial fibrillation, and hyperlipidemia (Table 2).

### 3.3. Cerebral Infarction Due to Thrombosis, Embolism, Occlusion, and Stenosis (CITO in ESKD Hospitalizations) 

There were 2,710,890 ESKD hospitalizations without CITO and 2304 ESKD hospitalizations with CITO (Table 3). In the cohort of hospitalized ESKD patients, those with CITO exhibited a higher average age compared to their counterparts without CITO (67.68 years vs. 61.34 years, *p* < 0.001, Table 3). In addition, those with CITO exhibited a higher average Elixhauser Comorbidity Index compared to those without CITO (6.01 vs. 5.33, *p* < 0.001, Table 3). Of the 2,710,890 ESKD hospitalizations without CITO, 66,890 had a KTx (2.47%), while of the 2304 ESKD hospitalizations with CITO, 37 had a KTx (1.61%) (*p*-value = 0.008). Of the 2,710,890 ESKD hospitalizations without CITO, 63,315 had PVD (2.34%), while of the 2304 ESKD hospitalizations with CITO, 96 had PVD (4.17%) (*p*-value < 0.001). Of the 2,710,890 ESKD hospitalizations without CITO, 1402 had a KTx and PVD (0.05%), while of the 2304 ESKD hospitalizations with CITO, 4 had a KTx and PVD (0.17%) (*p*-value = 0.033). 

In Table 4, the odds of hospitalization with CITO are presented across different dependent variables, risk factors, comorbidities, and demographic criteria. Between 2005 and 2019, KTx patients had 0.741 lower odds of being hospitalized with CITO compared to those with ESKD (CI 0.509–1.079, *p* = 0.118). Additionally, patients with ESKD and PVD had 1.332 higher odds of being hospitalized with CITO than ESKD patients without PVD (CI 1.074–1.653, *p* = 0.009). Meanwhile, patients with a KTx and PVD had 3.170 higher odds of being hospitalized with CITO than ESKD patients without KTx or PVD (CI 1.087–9.247, *p* = 0.035). 

### 3.4. Artery Occlusion and Stenosis Resulting in Cerebral Ischemia (AOSI in ESKD Hospitalizations)

There were 2,686,529 ESKD hospitalizations without AOSI and 26,665 ESKD hospitalizations with AOSI (Table 3). In the cohort of hospitalized ESKD patients, those with AOSI exhibited a higher average age compared to their counterparts without AOSI (70.43 years vs. 61.25 years, *p* < 0.001, Table 3). In addition, those with AOSI exhibited a higher average Elixhauser Comorbidity Index compared to those without AOSI (5.84 vs. 5.32, *p* < 0.001, Table 3). Of the 2,686,529 ESKD hospitalizations without AOSI, 66,556 had a KTx (2.48%), while of the 26,665 ESKD hospitalizations with AOSI, 371 had a KTx (1.39%) (*p*-value < 0.001). Of the 2,686,529 ESKD hospitalizations without AOSI, 61,467 had PVD (2.29%) while of the 26,665 ESKD hospitalizations with AOSI, 1944 had PVD (7.29%) (*p*-value < 0.001). Of the 2,686,529 ESKD hospitalizations without AOSI, 1381 had a KTx and PVD (0.05%) while of the 26,665 ESKD hospitalizations with AOSI, 25 had a KTx and PVD (0.09%) (*p*-value = 0.002).

In Table 4, the odds of hospitalization with AOSI are presented across different dependent variables, risk factors, comorbidities, and demographic criteria. Between 2005 and 2019, patients with a KTx had 0.760 lower odds of being hospitalized with AOSI than patients with ESKD (CI 0.678–0.851, *p* < 0001). Patients with ESKD and PVD had 2.626 higher odds of being hospitalized with AOSI than ESKD patients without PVD (CI 2.50–2.76, *p*-value < 0.001). Meanwhile, KTx patients with PVD had 0.983 lower odds of being hospitalized for AOSI than ESKD patients without a KTx or PVD (CI 0.649–1.49, *p*-value = 0.937).

### 3.5. Nontraumatic Intracranial Hemorrhage (NIH)

There were 2,686,118 ESKD hospitalizations without NIH and 17,076 ESKD hospitalizations with NIH (Table 3). In the cohort of hospitalized ESKD patients, those with NIH exhibited a higher average age compared to their counterparts without NIH (62.53 years vs. 61.33 years, *p* < 0.001, Table 3). In addition, those with NIH exhibited a higher average Elixhauser Comorbidity Index compared to those without NIH (5.63 vs. 5.32, *p* < 0.001, Table 3). Of the 2,686,118 ESKD hospitalizations without NIH, 66,532 (2.47%) had a KTx (2.47%), while of the 17,076 ESKD hospitalizations with NIH, 395 had a KTx (2.31%) (*p*-value = 0.194). Of the 2,686,118 ESKD hospitalizations without NIH, 63,131 had PVD (2.34%) while of the 17,076 ESKD hospitalizations with NIH, 280 had PVD (1.64%) (*p*-value < 0.001). Of the 2,686,118 ESKD hospitalizations without NIH, 1399 had a KTx and PVD (0.05%) while of the 17,076 ESKD hospitalizations with NIH, 7 had a KTx and PVD (0.04%) (*p*-value = 0.34).

In Table 4, the odds of hospitalization with NIH are presented across different dependent variables, risk factors, comorbidities, and demographic criteria. Between 2005 and 2019, patients with a KTx had 1.038 higher odds of being hospitalized with NIH compared than patients with ESKD (CI 0.932–1.157, *p* = 0.497). Patients with ESKD and PVD had 0.614 lower odds of being hospitalized with NIH than ESKD patients without PVD (CI 0.542–0.695, *p* < 0.001). Meanwhile, KTx patients with PVD had 1.041 higher odds of being hospitalized for NIH than ESKD patients without a KTx or PVD (CI 0.459–2.359, *p* = 0.924).

## 4. Discussion

In this study, we utilized the National Inpatient Sample database to investigate the rates of hospitalization for ischemic and hemorrhagic cerebrovascular diseases in patients with end-stage kidney disease (ESKD), peripheral vascular disease (PVD), kidney transplant (KTx), or a combination of these conditions. Our findings provide insights into the relationship between these comorbidities and the risk of cerebrovascular disease in this vulnerable patient population. 

First, we observed a lower percentage of hospitalized ESKD patients with CITO or AOSI who had undergone kidney transplantation compared to ESKD patients not hospitalized for CITO or AOSI. This finding suggests that patients with an increasing number of comorbidities may not qualify for or remain lower on the transplant list. Consequently, these patients experience prolonged exposure to the deleterious effects of hemodialysis, which contributes to the development of atherosclerotic disease and the subsequent risk of cerebral ischemia [27]. The lower hospitalization rates for strokes among kidney transplant recipients compared to ESKD patients support the previously reported protective role of kidney transplantation against the development of ischemic cerebrovascular disease [28,29].

Secondly, our findings revealed that ESKD patients with PVD exhibited increased likelihood of hospitalization due to CITO and AOSI, in contrast to those without PVD. Additionally, a greater proportion of ESKD patients hospitalized for CITO or AOSI showed the presence of PVD, in comparison to individuals not hospitalized with CITO or AOSI. These findings align with previous literature that demonstrated longer hospital stays and increased hospital costs for ESKD patients with concomitant PVD [9]. Given that ESKD patients face a three-fold increase in mortality risk after an ischemic stroke [30] compared to those without ESKD, the identification of ESKD patients with PVD as a high-risk population for ischemic cerebrovascular disease is of significant clinical importance [30].

Third, we observed a notably lower percentage of ESKD patients with PVD who were hospitalized with NIH compared to those without PVD (OR 0.641, 95% CI 0.542–0.695, *p* < 0.001). The coexistence of heightened cerebral ischemic events and decreased rates of hospitalization for cerebral hemorrhagic events in patients with ESKD and PVD elucidates unique questions regarding the proper management of these patients for the prevention of cerebrovascular events. Some studies propose that the heightened risk of ischemic events in ESKD patients, especially those with PVD, may be attributed to factors such as accelerated atherosclerosis, endothelial dysfunction, and thrombogenicity [1,8,29,31]. However, the observed lower rates of hospitalization for hemorrhagic events in this subgroup remain less explored. Recent research has indicated potential hemostatic alterations and platelet dysfunction in ESKD patients, possibly influencing the occurrence and severity of cerebral hemorrhagic events [31]. The delicate balance between prothrombotic and bleeding tendencies in this population adds complexity to the understanding of cerebrovascular complications. Moreover, considerations for anticoagulation and anti-platelet therapies in ESKD patients with PVD should be approached with caution due to the risk of bleeding complications [1,8,9]. Further research is warranted to elucidate the underlying mechanisms and inform targeted interventions to optimize the care and outcomes of this high-risk population.

When assessing cerebral ischemic versus hemorrhagic events in patients with both a kidney transplant and PVD, we revealed that within the subset of ESKD patients hospitalized for CITO, those who had both a kidney transplant and PVD showed a significantly heightened probability of being hospitalized (OR 3.17, 95% CI 1.087–9.247, *p* = 0.035). This is unexpected given that PVD alone only slightly increases the risk (OR 1.332, 1.074–1.6543, *p* = 0.009), while KTx alone has a minimal effect on hospitalization rates for CITO (OR 0.741, 0.509–1.079, *p* = 0.118). This suggests a strong correlation between the presence of a KTx, PVD, and the likelihood of hospitalization due to cerebral ischemic events. Furthermore, we noted that within the subset of ESKD patients hospitalized for NIH, those who had both a kidney transplant and PVD did not demonstrate a significantly heightened probability of being hospitalized specifically for NIH, rather than other reasons. 

At a surface level interpretation, these collective findings indicate a higher likelihood of hospitalization for cerebral ischemic events, rather than hemorrhage, among patients with both a KTx and PVD. Consequently, one could argue that optimal management for these individuals involves therapeutic anticoagulation, rather than withholding it to prevent potential adverse effects. Nevertheless, it is crucial to acknowledge that these conclusions are likely constrained by the inherent observational design of our study and the specific ICD-10 codes encompassed within its scope. Furthermore, it is worth noting that our study does not encompass periods of physiological stress or trauma, including the peri-transplant period [19].

The peri-transplant period is further complicated by the management of the patient’s anti-platelet therapy. Current guidelines recommend the use of anti-platelet therapy, such as aspirin or clopidogrel, for secondary prevention in patients with established cardiovascular disease, including stroke, in patients with symptomatic peripheral artery disease [11]. However, the evidence supporting the use of anti-platelet therapy specifically in patients with ESKD or kidney transplant and concurrent PVD is limited and conflicting [11]. The conflicting results may be attributed to the heterogeneity of study populations, variations in study designs, and differences in the definition of outcomes and follow-up duration. Additionally, the increased bleeding risk associated with anti-platelet therapy, especially in patients with ESKD or kidney transplant who may have impaired platelet function and increased susceptibility to bleeding complications, should be carefully considered. Further research is needed to clarify the role of anti-platelet therapy in the prevention of cerebrovascular events in patients with ESKD or kidney transplant and concurrent PVD. Prospective studies with larger sample sizes and longer follow-up periods are warranted to evaluate the efficacy and safety of different anti-platelet agents in this specific patient population. Additionally, studies investigating the underlying mechanisms of stroke in these patients, such as the contribution of platelet dysfunction, vascular calcification, and inflammation, may provide valuable insights for targeted preventions. In addition to medical management, aggressive control of traditional cardiovascular risk factors is crucial in reducing the risk of cerebrovascular events in patients with ESKD or KTx and concurrent PVD [5,6,14,18].

In tandem with appropriate patient management involving therapeutic anti-platelet therapy, it is essential to consider the impact of stroke post-renal transplant as a notable contributor to mortality [31]. Notably, the number of graft failures after transplant has been shown to be predictive of cerebrovascular events [28,31]. Moreover, PVD has been found to predict graft failure in kidney transplant patients as well [29,32]. Further investigation is warranted to understand the underlying pathophysiological characteristics and the relationship between PVD, graft failure, and the increased incidence of cerebrovascular events [29]. Evaluating this relationship may guide future interventions in the peri-transplant period to decrease mortality rates due to cerebrovascular diseases in these patients. 

Furthermore, close multidisciplinary collaboration between cardiologists, transplant surgeons, nephrologists, neurologists, and vascular surgeons is essential for the comprehensive management of patients with a kidney transplant and concurrent PVD. This collaboration can facilitate early identification, risk stratification, and appropriate interventions to reduce the burden of cerebrovascular disease in these patients.

Surprisingly, this study revealed that certain widely recognized risk factors for cerebrovascular disease exhibited negative associations [33,34,35]. Notably, obesity demonstrated a correlation with reduced odds of hospitalization for CITO, AOSI, and NIH. Essential hypertension was found to be linked with decreased odds of developing AOSI, while Type II diabetes mellitus was associated with a lower likelihood of hospitalizations for CITO and NIH. Pinpointing the unexpected inverse association observed in this study among well-known risk factors for cerebrovascular disease in patients with ESKD prompts consideration. Despite the robust existing evidence supporting the link between these risk factors and vascular disease, potential explanations may be rooted in diagnostic or temporal trends in coding practices over time [36]. Alternatively, it could signify a scenario of competing risks [37], suggesting that individuals with ESKD are more predisposed to encountering the detrimental effects of other vascular disease processes before experiencing cerebrovascular consequences [38]. Further investigation is necessary to unravel the complexities surrounding these unexpected associations.

Limitations of this study include its retrospective design which introduces inherent limitations such as selection bias, data accuracy, and coding which depend on the coding practices and completeness of the information recorded in the NIS database, and generalizability to other populations outside the United States. Other limitations include a lack of cause–effect relationship due to the observational nature of the study, potential unmeasured confounders, lack of clinical details that the NIS does not provide (such as PVD severity, kidney transplant outcomes, or specific treatments received), treatment and management variability across hospitals or physicians, potential residual confounding despite statistical adjustments, and lack of long-term outcomes such as mortality or functional outcomes. Despite these limitations, this nationwide analysis provides valuable insights into the rates of hospitalization for cerebrovascular diseases in patients with ESKD, PVD, and kidney transplants.

In summary, our study sheds light on hospitalization rates for both ischemic and hemorrhagic cerebrovascular diseases in patients dealing with ESKD, PVD, kidney transplants, or a combination of these factors. Notably, the presence of PVD in ESKD patients correlated with elevated hospitalization rates for cerebral ischemic events, while showing significantly decreased rates for hemorrhagic cerebral events. On the other hand, PVD presence in kidney transplant recipients was linked to notably increased hospitalization rates for ischemic cerebrovascular events, with no corresponding difference in hospitalization rates for hemorrhagic cerebral events. While not directly explored in this study, it is reasonable to speculate that the sustained rates of cerebral hemorrhage in patients with a kidney transplant and PVD, as opposed to the reduced rates seen in those with ESKD and PVD, could be influenced by factors related to the perioperative period. This observation underscores the need for a nuanced approach to managing patients with a kidney transplant and PVD, especially in the perioperative context.

## Figures and Tables

**Figure 1 healthcare-12-00454-f001:**
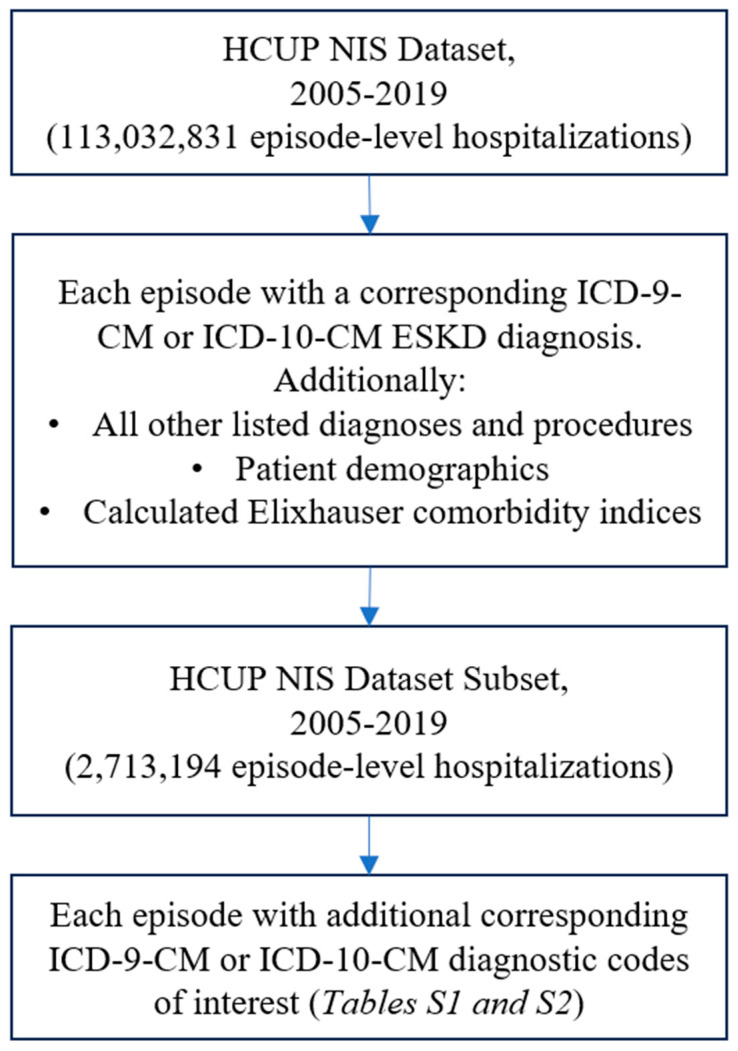
Depicts a PRISMA flow diagram representing the ESKD hospitalizations included in this study.

**Table 1 healthcare-12-00454-t001:** Descriptive Statistics, Baseline Characteristics, and Frequency Distribution of ESKD and Kidney Transplant Hospitalizations with and without PVD 2015–2019.

	ESKD without PVD (2,584,262)				ESKD with PVD (62,005)				KTx without PVD				KTx with PVD (1406)			
Key Conditions of Interest	*n* (%)	Range	Mean	Std. Dev.	*n* (%)	Range	Mean	Std. Dev.	*n* (%)	Range	Mean	Std. Dev.	*n* (%)	Range	Mean	Std. Dev.
Kidney Transplant	0				0				65,521 (100%)				1406 (100%)			
Peripheral Vascular Diseases	0				62,005 (100%)				0				1406 (100%)			
Kidney Transplant & Peripheral Vascular Diseases	0				0				0				1406 (100%)			
Cerebral infarction due to thrombosis, embolism, occlusion, and stenosis	2175 (0.08%)				92 (0.15%)				33 (0.05%)				4 (0.28%)			
Artery occlusion and stenosis resulting in cerebral ischemia	24,375 (0.94%)				1919 (3.09%)				346 (0.53%)				25 (1.78%)			
Nontraumatic intracranial hemorrhages	16,408 (0.63%)				273 (0.44%)				388 (0.59%)				7 (0.50%)			
** *Risk Factors and Comorbidities* **																
Essential Hypertension	31,612 (1.22%)				985 (1.59%)				1448 (2.21%)				29 (2.06%)			
Obesity	285,232 (11.04%)				8696 (14.02%)				4996 (7.63%)				158 (11.24%)			
Nicotine Dependence, Cigarettes	225,959 (8.74%)				8126 (13.11%)				4272 (6.52%)				115 (8.18%)			
Type II Diabetes Mellitus	571,459 (22.11%)				14,502 (23.39%)				10,449 (15.95%)				249 (17.71%)			
Atrial Fibrillation	371,734 (14.38%)				11,175 (18.02%)				7840 (11.97%)				255 (18.14%)			
Hyperlipidemia	748,491 (28.96%)				32,301 (52.09%)				17,621 (26.89%)				725 (51.56%)			
** *Other Specification Controls* **																
Elixhauser Comorbidity Index	2,584,262	1–18	5.30	1.98	62,005	1–18	6.89	1.98	65,521	1–16	4.69	1.94	1406	2–15	6.59	2.05
Age	2,584,082	0–114	61.44	15.78	62,002	8–99	66.25	12.31	65,517	0–97	52.78	15.65	1406	22–90	60.43	11.59
*Race*																
White	984,394 (38.09%)				28,498 (45.96%)				29,178 (44.53%)				670 (47.65%)			
Black	814,390 (31.51%)				19,564 (31.55%)				16,677 (25.45%)				380 (27.03%)			
Hispanic	379,976 (14.70%)				8657 (13.96%)				8712 (13.30%)				210 (14.94%)			
Asian or Pacific Islander	87,193 (3.37%)				1512 (2.44%)				2307 (3.52%)				31 (2.20%)			
Native American	26,389 (1.02%)				604 (0.97%)				510 (0.78%)				22 (1.56%)			
Other	69,913 (2.71%)				1385 (2.23%)				1754 (2.68%)				48 (3.41%)			
Missing	222,007 (8.59%)				1785 (2.88%)				6383 (9.74%)				45 (3.20%)			
** *Primary expected payer* **																
Medicare	1,913,716 (74.05%)				51,579 (83.19%)				48,181 (73.54%)				1192 (84.78%)			
Medicaid	297,103 (11.50%)				4547 (7.33%)				4841 (7.39%)				52 (3.70%)			
Private Insurance	287,101 (11.11%)				4539 (7.32%)				11,087 (16.92%)				140 (9.96%)			
Other	82,212 (3.18%)				1291 (2.08%)				1334 (2.04%)				21 (1.49%)			
Missing	4130 (0.16%)				49 (0.08%)				78 (0.12%)				1 (0.07%)			
** *Median household income national quartile for patient ZIP Code* **																
$1–$28,999	979,591 (37.91%)				24,150 (38.95%)				19,997 (30.52%)				444 (31.58%)			
$29,000–$35,999	633,972 (24.53%)				15,282 (24.65%)				16,478 (25.15%)				343 (24.40%)			
$36,000–$46,999	528,582 (20.45%)				12,655 (20.41%)				15,259 (23.29%)				325 (23.12%)			
$47,000+	385,058 (14.90%)				8858 (14.29%)				12,485 (19.05%)				267 (18.99%)			
Missing	57,059 (2.21%)				1060 (1.71%)				1302 (1.99%)				27 (1.92%)			
** *Patient Location: NCHS Urban-Rural Code* **																
Central counties of metro areas of ≥1 million population	1,001,363 (38.75%)				21,065 (33.97%)				22,665 (34.59%)				465 (33.07%)			
Fringe counties of metro areas of ≥1 million population	565,413 (21.88%)				13,954 (22.50%)				15,955 (24.35%)				355 (25.25%)			
Counties in metro areas of 250,000–999,999 population	452,963 (17.53%)				12,630 (20.37%)				12,158 (18.56%)				268 (19.06%)			
Counties in metro areas of 50,000–249,999 population	204,123 (7.90%)				5202 (8.39%)				5589 (8.53%)				114 (8.11%)			
Micropolitan counties	202,325 (7.83%)				5127 (8.27%)				5121 (7.82%)				113 (8.04%)			
Not metropolitan or micropolitan counties	130,219 (5.04%)				3781 (6.10%)				3326 (5.08%)				82 (5.83%)			
Missing	27,856 (1.08%)				246 (0.40%)				707 (1.08%)				9 (0.64%)			
1 Dataset Contains Only ESKD Patients (*n* = 2,713,194)																

**Table 2 healthcare-12-00454-t002:** Baseline Characteristics for all ESKD patients with PVD versus Patients without PVD2005-2019. Key conditions of interest, risk factors and comorbidities, and other specifications controls utilize *t*-tests to evaluate differences between groups. Race, primary expected payer, median household income, and patient location utilize chi-squared analysis.

Key Condition Patient Differences (*t*-Test & Chisq)	Peripheral Vascular Diseases		
Key Conditions of Interest	No Condition (*n* = 2,649,783)	Condition (*n* = 63,411)	*p*-Value
Kidney Transplant, *n* (%)	65,521 (2.47%)	1406 (2.22%)	<0.001
Cerebral infarction due to thrombosis, embolism, occlusion, and stenosis, *n* (%)	2208 (0.08%)	96 (0.15%)	<0.001
Artery occlusion and stenosis resulting in cerebral ischemia, *n* (%)	24,721 (0.93%)	1944 (3.07%)	<0.001
Nontraumatic intracranial hemorrhages, *n* (%)	16,796 (0.63%)	280 (0.44%)	<0.001
** *Risk Factors and Comorbidities* **			
Essential Hypertension	33,060 (1.25%)	1014 (1.60%)	<0.001
Obesity	290,228 (10.95%)	8854 (13.96%)	<0.001
Nicotine Dependence, Cigarettes	230,231 (8.69%)	8241 (13%)	<0.001
Type II Diabetes Mellitus	581,908 (21.96%)	14,751 (23.26%)	<0.001
Atrial Fibrillation	379,574 (14.32%)	11,430 (18.03%)	<0.001
Hyperlipidemia	766,112 (28.91%)	33,026 (52.08%)	<0.001
** *Other Specification Controls* **			
Elixhauser Comorbidity Index (1–18), mean (SD)	5.29 (1.98)	6.88 (1.98)	<0.001
Age, mean (SD)	61.23 (15.83)	66.12 (12.32)	<0.001
*Race, n (%)*			
White	1,013,572 (41.86%)	29,168 (47.37%)	<0.001
Black	831,067 (34.32%)	19,944 (32.39%)	
Hispanic	388,688 (16.05%)	8867 (14.40%)	
Asian or Pacific Islander	89,500 (3.70%)	1543 (2.51%)	
Native American	26,899 (1.11%)	626 (1.02%)	
Other	71,667 (2.96%	1433 (2.33%)	
** *Primary expected payer, n (%)* **			
Medicare	1,961,897 (74.16%)	52,771 (83.29%)	<0.001
Medicaid	301,944 (11.41%)	4599 (7.26%)	
Private Insurance	298,188 (11.27%)	4679 (7.38%)	
Other	83,546 (3.16%)	1312 (2.07%)	
** *Median household income national quartile for patient ZIP Code, n (%)* **			
$1–$28,999	999,588 (38.57%)	24,564 (39.46%)	<0.001
$29,000–$35,999	650,450 (25.10%)	15,625 (25.07%)	
$36,000–$46,999	543,841 (20.99%)	12,980 (20.83%)	
$47,000+	397,543 (15.34%)	9125 (14.64%)	
** *Patient Location: NCHS Urban-Rural Code, n (%)* **			
Central counties of metro areas of ≥1 million population	1,024,028 (39.07%)	21,530 (34.09%)	<0.001
Fringe counties of metro areas of ≥1 million population	581,368 (22.18%)	14,309 (22.66%)	
Counties in metro areas of 250,000–999,999 population	465,121 (17.74%)	12,898 (20.42%)	
Counties in metro areas of 50,000–249,999 population	209,712 (8.00%)	5316 (8.42%)	
Micropolitan counties	207,446 (7.91%)	5240 (8.30%)	
Not metropolitan or micropolitan counties	133,545 (5.09%)	3863 (6.12%)	

**Table 3 healthcare-12-00454-t003:** *t*-test results comparing the rates of hospitalization for cerebrovascular diseases amongst patients with key conditions of interest from 2005 to 2019.

	Hospitalization for Cerebral Infarction Due to Thrombosis, Embolism, Occlusion, and Stenosis (CITO)			Hospitalization for Artery Occlusion and Stenosis Resulting in Cerebral Ischemia (AOSI)			Hospitalization for Nontraumatic Intracranial Hemorrhage (NIH)		
Key Conditions of Interest **	Not Hospitalized with CITO (*n* = 2,710,890)	Hospitalized with CITO (*n* = 2304)	*p*-Value	Not Hospitalized with AOSI (*n* = 2,686,529)	Hospitalized with AOSI (*n* = 26,665)	*p*-Value	Not Hospitalized with NIH (*n* = 2,696,118)	Hospitalized with NIH (*n* = 17,076)	*p*-Value
Kidney Transplant, *n* (%)	66,890 (2.47%)	37 (1.61%)	0.008	66,556 (2.48%)	371 (1.39%)	<0.001	66,532 (2.47%)	395(2.31%)	0.194
ESKD & Peripheral Vascular Diseases, *n* (%)	63,315 (2.34%)	96 (4.17%)	<0.001	61,467 (2.29%)	1944 (7.29%)	<0.001	63,131 (2.34%)	280 (1.64%)	<0.001
Kidney Transplant & Peripheral Vascular Diseases, *n* (%)	1402 (0.05%)	4 (0.17%)	0.033	1381 (0.05%)	25 (0.09%)	0.002	1399 (0.05%)	7 (0.04%)	0.34
**Other Specification Controls**									
Age, mean (SD)	61.34 (15.78)	67.68 (12.03)	<0.001	61.25 (15.79)	70.43 (10.82)	<0.001	5.32 (1.99)	5.63 (2.06)	<0.001
Elixhauser Comorbidity Index (1–18), mean (SD)	5.33 (1.99)	6.01 (2.06)	<0.001	5.32 (1.99)	5.84 (2.02)	<0.001	61.33 (15.78)	62.53 (14.74)	<0.001

* Fisher’s exact test, ** Data set only contains ESKD patients (*n* = 2,713,194).

**Table 4 healthcare-12-00454-t004:** Displays key regression specifications including comorbidities, risk factors, race, primary expected payer, median household income, and patient location amongst ESKD patients.

Key Regression Specifications						
Dependent Variable	CITO		AOSI		NIH	
	OR (95% CI)	*p*-Value	OR (95% CI)	*p*-Value	OR (95% CI)	*p*-Value
KiTx ^1^	0.741 (0.509–1.079)	0.118	0.76 (0.678–0.851)	<0.001 *	1.013 (0.932–1.157)	0.814

Peripheral Vascular Diseases	1.332 (1.074–1.653)	0.009 *	2.626 (2.50–2.76)	<0.001 *	0.641 (0.542–0.695)	<0.001 *

KTx & Peripheral Vascular Diseases	3.17 (1.087–9.247)	0.035 *	0.983 (0.649–1.49)	0.937	1.040 (0.459–2.358)	0.925
** *Risk Factors and Comorbidities* **						
Essential Hypertension	0.745 (0.468–1.186)	0.215	0.865 (0.763–0.981)	0.024 *	1.089 (0.946–1.253)	0.236

Obesity	0.783 (0.678–0.905)	0.001 *	0.906 (0.869–0.945)	<0.001 *	0.471 (0.440–0.503)	<0.001 *

Nicotine Dependence, Cigarettes	0.9 (0.757–1.07)	0.233	1.298 (1.239–1.360)	<0.001 *	0.725 (0.678–0.774)	<0.001 *

Type II Diabetes Mellitus	0.869 (0.781–0.966)	0.009 *	1.096 (1.065–1.129)	<0.001 *	0.931 (0.895–0.968)	<0.001 *

Atrial Fibrillation	0.896 (0.798–1.007)	0.066	0.906 (0.876–0.937)	<0.001 *	0.821 (0.782–0.862)	<0.001 *

Hyperlipidemia	1.301 (1.188–1.424)	<0.001 *	1.953 (1.903–2.005)	<0.001 *	0.864 (0.736–0.793)	<0.001 *
** *Other Specification Controls* **						
Elixhauser Comorbidity Index (1–18)	1.146 (1.132–1.178)	<0.001 *	1.054 (1.047–1.061)	<0.001 *	1.119 (1.110–1.128)	<0.001 *

Age	1.022 (1.019–1.025)	<0.001 *	1.038 (1.037–1.039)	<0.001 *	1.005 (1.004–1.006)	<0.001 *
** *Race* **						
Race (Black) ^2^	0.884 (0.790–0.986)	0.029 *	0.595 (0.575–0.616)	<0.001 *	1.206 (1.157–1.257)	<0.001 *

Race (Hispanic) ^2^	0.863 (0.735–0.973)	0.038 *	0.73 (0.700–0.762)	<0.001 *	1.260 (1.199–1.324)	<0.001 *

Race (Asian or Pacific Islander) ^2^	0.836 (0.637–1.026)	0.14	0.76 (0.710–0.813)	<0.001 *	1.932 (1.802–2.072)	<0.001 *

Race (Native American) ^2^	0.497 (0.263–0.921)	0.029 *	0.932 (0.819–1.061)	0.286	1.159 (0.983–1.368)	0.08

Race (Other) ^2^	0.865 (0.647–1.131)	0.31	0.811 (0.747–0.879)	<0.001 *	1.486 (1.361–1.622)	<0.001 *
** *Primary Expected Payer* **						
Primary expected payer (Medicaid) ^3^	0.737 (0.609–0.893)	0.001 *	0.683 (0.641–0.729)	<0.001 *	1.022 (0.967–1.079)	0.441

Primary expected payer (Private Insurance) ^3^	1.097 (0.944–1.269)	0.218	0.991 (0.946–1.038)	0.69	1.214 (1.153–1.277)	<0.001 *

Primary expected payer (Other) ^3^	0.98 (0.738–1.298)	0.889	0.67 (0.604–0.742)	<0.001 *	1.259 (1.154–1.373)	<0.001 *
** *Median Household Income* **						
Median household income national quartile for patient ZIP Code ($29,000–$35,999) ^4^	1.031 (0.916–1.152)	0.597	1.076 (1.040–1.114)	<0.001 *	1.013 (0.971–1.057)	0.557

Median household income national quartile for patient ZIP Code ($36,000–$46,999) ^4^	0.951 (0.832–1.074)	0.442	1.089 (1.049–1.130)	<0.001 *	1 (0.955–1.048)	0.987

Median household income national quartile for patient ZIP Code ($47,000+) ^4^	1.073 (0.925–1.223)	0.323	1.099 (1.054–1.146)	<0.001 *	1.047 (0.993–1.103)	0.09
** *Patient Location* **						
Patient location: fringe counties of metro areas of ≥1 million population ^5^	0.892 (0.790–1.004)	0.061	1.063 (1.026–1.102)	0.001 *	0.933 (0.893–0.975)	0.002 *

Patient location: counties in metro areas of 250,000–999,999 population ^5^	0.961 (0.843–1.085)	0.541	1.149 (1.106–1.192)	<0.001 *	1.023 (0.977–1.070)	0.336

Patient location: counties in metro areas of 50,000–249,999 population ^5^	0.842 (0.701–1.008)	0.064	1.162 (1.105–1.221)	<0.001 *	0.944 (0.884–1.008)	0.087

Patient location: micropolitan counties ^5^	1 (0.841–1.197)	0.999	1.14 (1.083–1.201)	<0.001 *	0.932 (0.870–0.999)	0.048 *

Patient location: not metropolitan or micropolitan counties ^5^	1.058 (0.865–1.299)	0.586	1.297 (1.224–1.375)	<0.001 *	0.971 (0.894–1.054)	0.48
Observations	2,408,114		2,408,114		2,408,114	

^1^ Reference group: ESKD. ^2^ Reference group: White. ^3^ Reference group: Medicare. ^4^ Reference group: Median household income national quartile for patient ZIP Code ($1–$28,999). ^5^ Reference Group: Patient location: central counties of metro areas of ≥1 million population. * statistical significant finding using *p* < 0.05.

## Data Availability

All data included in this study are available via the publicly available HCUP database.

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
