# Peer review of "Cerebrovascular Disease Hospitalization Rates in End-Stage Kidney Disease Patients with Kidney Transplant and Peripheral Vascular Disease: Analysis Using the National Inpatient Sample (2005–2019)"

_healthcare, 2024, doi:10.3390/healthcare12040454_

Round 1

Reviewer 1 Report

Comments and Suggestions for Authors

The study explored the impact of peripheral vascular disease (PVD) and kidney transplantation (KiTx) on hospitalization rates for cerebral infarction and hemorrhage. Using 2,713,194 end-stage renal disease (ESRD) hospitalizations (2005-2019) using the National Inpatient Sample, we investigated hospitalization rates for ischemic and hemorrhagic cerebrovascular diseases concerning ESRD, PVD, KiTx, or their combinations. This nationwide analysis reveals that PVD in ESRD patients is associated with increased hospitalization rates with cerebral ischemic events and reduced NIH events. Among KiTx recipients, PVD correlated with increased hospitalizations for ischemic events, without affecting NIH. This highlights management concerns for patients with KiTx and PVD.

Major Comments

·       Retrospective descriptive analysis and multivariate logistic regressions were performed to examine how the probability of each condition of interest vary between ESRD patients, kidney transplant recipients, and if they additionally had PVD. Descriptive analysis was done using a t-test for continuous variables and Chi-squared or Fishers exact tests for categorical variables, depending on the sample size and the distribution of the variables

·       From Tables 2-3 or Figure 1, It is hard to find which test resulted in which p values as well as the output of multivariate logistic regressions.

·       Patients hospitalized with NIH showed similar rates of KiTx and PVD (0.04%) versus non-NIH hospitalizations (0.05%,p=0.34). The P value>0.05. Is this highly significant?

without a 174 KiTx or PVD (CI 0.649-1.49, p-value=0.937)

·       Authors should mention the P value significance cut-off so that the readers will get a better understanding.

·       For Table 2 some values are not formatted correctly

Overall Comments

The study findings provide insights into the relationship between the above-mentioned comorbidities and the risk of cerebrovascular disease in this vulnerable patient population.

The three major findings and interpretations are valuable to the researchers to address the significant research problems.

In summary, the study sheds light on hospitalization rates for both ischemic and hemorrhagic cerebrovascular diseases in patients dealing with ESRD, PVD, kidney transplants, or a combination of these factors. Notably, the presence of PVD in ESRD patients correlated with elevated hospitalization rates for cerebral ischemic events, while showing significantly decreased rates for hemorrhagic cerebral events.

Accept with minor revision.

Reviewer 2 Report

Comments and Suggestions for Authors

This study explores the relationship between the risk of peripheral vascular disease (PVD) and various types of cerebrovascular diseases in patients with end-stage kidney disease (ESKD) and kidney transplant recipients (KiTx). While the study is intriguing, several concerns need addressing:

  1. 1. The study design would be more robust if conducted as a retrospective cohort study instead of a case-control study. Evaluating the association of PVD and cerebrovascular diseases in ESKD and KiTx would be more meaningful by identifying groups with and without PVD and then comparing the rates of cerebrovascular diseases (CITO, AOSI, NIH) in each group. The current case-control approach, assessing outcomes first and then identifying exposures, may lead to bias and does not adequately support the conclusion that PVD is an associated risk factor for the ESKD/KT population. For example, this approach has the potential for misinterpretation, as it includes patients hospitalized for reasons other than CITO but diagnosed with AOSI and NIH, patients hospitalized for reasons other than AOSI but included with CITO and NIH, and patients hospitalized for reasons other than NIH but included with CITO and AOSI. This may introduce confounding factors and compromise the validity of the study findings.

  2.  
  3. 2. Descriptive statistics in Table 2 should present and compare baseline characteristics in each of the four groups: KiTx without PVD, KiTx with PVD, ESKD with PVD, and ESKD without PVD. This would allow a better examination of whether baseline patient characteristics are balanced between these groups.

  4.  
  5. 3. Additional baseline characteristics associated with cerebrovascular disease, such as age, sex, and underlying diseases (e.g., diabetes, hypertension, atrial fibrillation, cardiovascular diseases), should be evaluated and reported. Further regression analysis should be conducted to determine if PVD remains a significant factor for cerebrovascular diseases after adjusting for these risk factors.

  6.  
  7. 4. It is recommended to include a flow study diagram illustrating the numbers of included and excluded study populations at each step. This graphical representation, presented as Figure 1, will enhance clarity regarding the number of analyzed populations in each group.

  8.  
  9. 5. Clarify the distinction between KiTx (n=66,927) and KiTx with PVD (n=1,406). Does KiTx refer to KiTx without PVD, or does it encompass both KiTx without PVD and KiTx with PVD?

  10.  
  11. 6. Inclusion and exclusion criteria should be clearly outlined in the Methods section. Specify whether the study includes pediatric and pregnant populations or patients with known cases of cerebrovascular disease and readmission.

  12.  
  13. 7. Clearly state the study outcomes and objectives in the Methods section.

  14. 8. Consider using "End-stage kidney disease (ESKD)" rather than "End-stage renal disease (ESRD)" in the title and throughout the manuscript for consistency.

Reviewer 3 Report

Comments and Suggestions for Authors

In the manuscript titled "Cerebrovascular Disease Hospitalization Rates in ESRD Patients with Kidney Transplant & PVD: Analysis using the National Inpatient Sample (2005-2019)," the authors assessed hospitalization rates for various cerebral infarction or hemorrhage events in patients with ESRD, PVD, kidney transplant, or a combination of these conditions by analyzing the National Inpatient Sample (NIS) from the USA. To bolster the findings, please consider incorporating the following comments into your manuscript before proceeding with the next steps in the review process. Your attention to these suggestions will contribute to the overall enhancement of your work, and we appreciate your dedication to refining your article. 

Major comments

1)      Some clinical variables, such as systemic arterial hypertension (SAH), diabetes mellitus (DM), and smoking, were not included in the analyses. These factors contribute to the burden of ESRD, posing a significant risk for cerebrovascular diseases, especially in older individuals and those with a longer time on dialysis. The etiology of ESRD (SAH, DM, glomerulonephritis, ADPKD, among others), a crucial factor in increasing morbidity, has not been analyzed or discussed. Similarly, smoking, a well-known risk for PVD and cardio- and cerebrovascular diseases, has not been addressed.

2)      Consequently, the outcomes should be stratified based on the time of ESRD and kidney transplant (KTx), etiology, and associated comorbidities of ESRD individuals and KTx recipients, as well as age and sex.

3)      Addressing the aforementioned topics would provide a comprehensive view of the impact of PVD and renal disease on cerebrovascular diseases.

Minor comment

1)      On page 3, in section 2.3 Analysis, please include the significance level ("P" value) considered when performing statistical tests.

Round 2

Reviewer 2 Report

Comments and Suggestions for Authors

I appreciate the effort of the authors to put into the revised manuscript. However, I would recommend making some improvement:

  1. - Table 1 could benefit from refinement. I suggest streamlining it by removing the "range, mean, std dev" column. This information could be incorporated into the "N(%)" column or denoted with asterisks (and presenting this information below the table).

  2.  
  3. - Supplement Tables 1 and 2 should be relocated to the supplemental materials section, rather than being included in the main manuscript.

  4.  
  5. - Regarding Table 4, I find it intriguing. However,

  6. it would enhance the manuscript to include explanations in the discussion for the following points:
  7. 1. Why do patients with PVD exhibit a lower risk of admission due to NIH compared to those without PVD (OR 0.641, 95% CI 0.542-0.695)?

    2. Why does the combination of KTx & PVD significantly increase the risk of admission due to CITO (OR 3.17)? This is unexpected given that PVD alone only slightly increases the risk (OR 1.13), while KTx alone decreases it (OR 0.741).

    3. Discuss the unexpected outcomes of other risk factors, such as essential HT decreasing the risk of admission due to AOSI, obesity decreasing the risk of admission due to CITO/AOSI/NIH, T2DM significantly decreasing the risk of admission due to CITO and NIH, AF decreasing the risk of admission due to AOSI/NIH, and hyperlipidemia decreasing the risk of admission due to NIH. Explain why these findings deviate from current knowledge.

Overall, the manuscript has improved, and addressing these suggestions will further enhance its quality.

Reviewer 3 Report

Comments and Suggestions for Authors

No further comments. Congratulations on your work. 

Author Response

Thank you so much for your help in improving our manuscript!